# Psychological Trust Dynamics in Climate Change Adaptation Decision-Making Processes: A Literature Review

**Rubinia Celeste Bonfanti [1]**, **Stefano Ruggieri [2,*]** and **Adriano Schimmenti [1]**

[1] Faculty of Human and Social Sciences, Kore University of Enna, 94100 Enna, Italy; rubiniaceleste.bonfanti@unikore.it (R.C.B.); adriano.schimmenti@unikore.it (A.S.)
[2] Department of Psychology, Educational Science and Human Movement, University of Palermo, 90128 Palermo, Italy
* Correspondence: stefano.ruggieri@unipa.it

**Abstract:** There has been a growth in interest among academics and professionals in psychological trust dynamics during climate change adaptation. This literature review aimed to examine the research concerning trust dynamics in climate change adaptation from different levels of analysis, encompassing the different phases of adaptation and considering the importance of trust in climate change decision-making. The method consisted of systematically reviewing researches on this topic published in scientific articles, by using appropriate and relevant search keywords (e.g., trust, community, natural hazard, climate change adaptation, decision-making) in academic databases. A total of 25 studies met inclusion criteria. All the articles focused on the latter phases of the climate change adaptation cycle, specifically implementation and monitoring/evaluation, with limited attention devoted to decision-making related to earlier phases of preparation, assessment of risks, and identification and selection of adaptation options. The reviews also indicates that psychological trust is related to different adaptive actions (e.g., adoption of renewable energy technologies), and low- and high-impact mitigation behaviors (e.g., acceptance of paying taxes for actions that mitigate climate change). Therefore, this review underscores the significant role of psychological trust dynamics in shaping individuals' decision-making processes concerning climate change adaptation, thereby yielding immediate and direct implications for climate systems. Therefore, it is essential to actively promote the culture of trust within the context of climate change adaptation.

**Keywords:** trust; climate change adaptation; climate-related hazards; climate change decision-making

## 1. Introduction

In recent years, researchers have shown a growing interest in examining the role of psychological trust dynamics in climate change adaptation (CCA). Academics and professionals across various fields, encompassing economics, law, psychology, and the social sciences, have acknowledged trust as a pivotal determinant impacting the extent of community responsiveness in the context of climate-related hazards [1,2]. This happens because the process of adjusting to climate change demands changes in attitudes and behaviors among millions of individuals. Notably, individuals and communities make both individual and collective choices that are significantly influenced by climate-related events and have the potential to impact the planet's climate equilibrium [2,3].

Trust encompasses a broad definition as both a mental condition involving a willingness to embrace vulnerability stemming from optimistic beliefs regarding the intentions, or the actions of another individual or group [4]. Trust is pivotal in the interpersonal dynamics among individuals and societies. Indeed, trust requires taking a leap of faith, wherein the individual or group extending trust willingly embraces vulnerability in their connection with the trusted entity, whether it be a person, a group, or an institution. This involves receiving explicit assurances or guarantees regarding the underlying intentions and actions of the trusted party [5,6].

According to Walker and Hills [7], trust constitutes a pivotal element in the establishment and sustenance of connections among individuals who might not engage with one another under different circumstances. Within this framework, trust can be regarded as a relational construct delineating the quality of interpersonal communication among individuals, groups, organizations, and so on. It allows individuals to build connections within their social environment and institutions, thereby promoting the mutual exchange of support and assistance [8]. Henceforth, trust is intricately associated with the notion of community. Precisely, community trust can be delineated as a fundamental attribute encompassing positive attitudes toward others, confidence in active participation, and the assurance that the community can amicably address divergences while engaging in collectively endorsed public endeavors [9]. Furthermore, trust shows potential in alleviating the adverse impacts of psychological distress faced by marginalized or economically disadvantaged communities [7,10].

Moreover, in recent times, there has been a growing interest in investigating trust in government and institutions. The concept of institutional trust revolves around the assurance in the competence of institutions to tackle diverse risks and social issues [11]. This concept is frequently linked to the anticipation that institutions will enact policies that are advantageous and efficacious for citizens [7,10]. A communities' evaluation of an institutions' implement policies becomes especially crucial when risks and costs emanate from sources primarily beyond the control of individuals. The institution bears the responsibility of foreseeing and dealing with damages that largely exceed the control of individuals, as it happens in the contexts of disaster risk reduction and recovery after natural hazards.

The relevance of trust in reducing complications and costs resides in cultivating a resilient sense of care, solidarity, and engaged involvement within the community. This, in turn, may enhance the community's ability to respond more effectively to emergencies [3,12].

Therefore, the concept of trust also encompasses an instrumental dimension concerning the outcomes of interactions between citizens and institutions. Notably, institutional trust is susceptible to various factors, including individual knowledge and the perceived competence of emergency personnel [13]. Aligned with the concept of individual knowledge, the enhancement of public knowledge could serve as a potential instrument to elevate public trust in climate-related hazards management, thus contributing to effective extreme events preparedness plans [14]. Moreover, the perceived competency of climate-related hazards professionals, characterized by the belief in their capacity to formulate proactive plans against extreme events, an active involvement with the community, and thus ensuring community resilience, plays a crucial role in cultivating trust in institutions [14]. In instances where the processes of predicting and addressing damages are inadequately managed, there exists the potential to undermine the confidence of extreme events victims in the fairness and justice of society [15]. This, in turn, can lead to a widespread erosion of trust in institutions, suggesting a perception that institutions cannot be relied upon to provide necessary resources or to act to ensure safety and justice in times of adversity. Consequently, the enduring perception of threat may persist over time, contributing to the maintenance of psychological distress within the community [16].

In the domain of behaviors characterized by both low and high environmental impacts, another level of trust has been identified, which is scientific-technological trust. Trust in science and technology stands as a fundamental concept within the technological or scientific domain, often eluding the awareness of numerous researchers [17]. It constitutes a variant of community and institutional trust, characterized by the attribution of impersonal trust to individuals engaged in scientific pursuits and technological enterprises within institutional contexts (such as scientists or technological experts) [18]. Science, technology, and trust represent three integral components, both tangible and abstract, inextricably linked to each other. Individuals characterized by a disposition of distrust are resistant to embracing technology and science, and if this resistance remains unchecked, it has the potential to manifest as a global phenomenon [19]. Specifically, the adoption of technology is contingent upon cultivating trust in its positive and substantial impacts through fostering

an appropriate mindset. Consequently, a positive association exists between the level of trust and the corresponding attitude towards technology [20]. Indeed, there is evidence that scientific-technological trust is positively associated with the uptake in both public and private pro-environmental behaviors [17,21] as well as collective action on climate change [22]. Indeed, this type of trust can function as a heuristic in the decision-making process, aiding individuals in navigating and responding to complexities within their environments [23,24].

Therefore, scientific-technological trust emerges as a crucial catalyst for fostering collaboration among stakeholders, a dynamic cultivated through repeated interactions between the involved parties [25]. Moreover, it plays a pivotal role in augmenting the inclination to financially support restoration endeavors [17,26,27]. According to the findings of Cologna and Siegrist [17], reported a positive and robust association between trust in scientists and the engagement in climate-friendly behaviors. This observation is further supported by Tranter and Lester [28] who determined that individuals who exhibit trust in scientists as purveyors of environmental information tend to prioritize initiatives addressing climate change. Moreover, research by Leiserowitz et al. [29] and Tranter and Lester [28] underscore that heightened trust in climate scientists and the scientific enterprise as a whole corresponds to an increased likelihood of acknowledging human-induced factors in climate change. The impact of technology extends indirectly to public engagement, either by mitigating erroneous perceptions of environmental risks or by enhancing beliefs in the benefits of environmental interventions [26]. Consequently, the absence of scientific-technological trust constitutes a foundational impediment to the planning and execution of restoration initiatives [25].

In light of the preceding considerations and the pivotal significance of trust in the context of CCA, it is imperative to highlight the variegated psychological trust dynamics within communities, institutions, and scientific spheres, particularly as they differ between developing and developed nations. Within developed industrialized societies, individuals engage in a broader spectrum of societal affiliations, wherein dependence on community trust is notably diminished, supplanted instead by trust in institutional mechanisms overseeing communal governance on a larger scale. Conversely, in developing countries where localized communities wield substantial influence over individual livelihoods, the bedrock of social capital primarily rests upon community trust. Nevertheless, in the absence of complementary forms of trust, this reliance on community trust may foster insular, tightly cohesive communities, hindering broader involvement with developmental perspectives, efficiency, and innovation—including trust in scientific and technological domains—across more expansive, diverse groups. Within this framework, developing nations, characterized by less developed formal institutional infrastructures, are characterized by lower levels of institutional trust compared to their developed counterparts [30].

The main aims of this literature review are to analyze and summarize the outcomes of diverse studies on the role of trust in the different phases of CCA, and to understand how trust dynamics are linked to individuals' climate change decision-making.

### 1.1. Psychological Trust Dynamics, Climate Change Adaptation, and Climate Change Decision-Making Process

The frequency, intensity, and impact of climate change and its associated disasters like storms, floods, rising sea levels, cyclones, and droughts are projected to escalate [31,32]. Fifty percent of the global population currently resides in regions susceptible to climate-related disasters [32] and the repercussions of climate change may exhibit a cascading nature, potentially resulting in adverse effects on health, malnutrition, migration, and social conflicts.

The necessity of integrating measures to adapt to climate change and mitigate climate-related disasters is increasingly acknowledged within policy frameworks. The pivotal role of communities and local actors in promoting CCA and resilience to disasters is considered crucial in these responsive initiatives [33]. The definition of CCA in human systems, as

outlined by the United Nations Intergovernmental Panel on Climate Change, involves "the process of adjustment to actual or expected climate and its effects, in order to moderate harm or exploit beneficial opportunities" [32]. The United Nations Development Program (UNDP) presents adaptation as "a process by which strategies to moderate, cope with and take advantage of the consequences of climate events are enhanced, developed and implemented" [34]. Also, Berrang-Ford et al. [35] examine the adaptation "endeavors", which encompass what a government actively undertakes in response to its vulnerabilities and adaptation objectives, as well as how governments communicate, mobilize, and coordinate for adaptation.

The CCA cycle consists of six steps [36]: (1) preparing the ground for adaptation, (2) assessing climate change risks and vulnerabilities, (3) identifying adaptation options, (4) assessing and selecting adaptation options, (5) implementing adaptation, and (6) monitoring and evaluating adaptation.

The first step (preparing the ground for adaptation) introduces the key elements required to start the adaptation process by creating a favorable political situation for adaptation and identifying evidence and data on current and potential future climate impacts, adaptation actions, and examples of good practices. The second step (assessing climate change risks and vulnerabilities) outlines the methodology for evaluating the impact of climate change on individuals, sectors, or systems, contingent upon three distinct factors: climate-related hazards, vulnerability, and exposure; although climate-related hazards are directly influenced by climate change, vulnerability and exposure are also contingent upon socio-economic factors. The third step (identifying adaptation options) has the goal of identifying a set of adaptation options to tackle the climate challenges identified in the previous step. The fourth step (assessing and selecting adaptation options) requires a close collaboration with expert stakeholders for assessing and prioritizing the potential adaptation options. In the fifth step (implementing adaptation) policies necessitate implementation across a given locality through the formulation and execution of an adaptation strategy and action plan under the purview of regional or local authorities. The last step (monitoring and evaluating adaptation) assists in assessing the effectiveness of adaptation measures and identifying any unforeseen side effects that may have occurred.

The process of adaptation also encompasses a varied range of actions, classified as structural, institutional, ecological, or behavioral. Structural adaptations entail the physical alteration of infrastructure and the built environment, such as the construction of flood barriers. Institutional adaptations involve creating policies, regulations, and governance mechanisms to facilitate adaptive responses, such as establishing a climate change adaptation department within a government agency. Ecological adaptations center on restoring and conserving natural ecosystems to bolster their resilience to climate impacts, such as creating wetlands for natural flood protection, enhanced biodiversity, and carbon sequestration. Lastly, behavioral adaptations encompass alterations in individual and collective behaviors, practices, and decision-making processes, for instance, the adoption of water conservation practices by households, the adoption of renewable energy technologies, or the willingness to engage in both low- and high-impact mitigation behaviors [37–39].

Recent studies underscore the pivotal role that cities play in tackling climate change by both mitigating its effects and adapting to them. Take, for instance, spatial planning, which is increasingly acknowledged as a fundamental tool. It facilitates the integration of urban design, the optimization of spatial structure, and the efficient management of land use and infrastructure. Moreover, it acts as a governance mechanism at the local level, shaping policy measures for spatial development. In recent years, western nations have incorporated climate adaptation objectives into their spatial planning policies, while eastern countries have been involved in climate programs since the late 1980s. Recent revisions in territorial and spatial planning aim to holistically coordinate various spatial aspects. Nevertheless, there is a pressing need to strengthen the role of territorial planning in addressing climate change at the urban level, especially due to inadequate awareness, limited analytical capability, and insufficient action regarding climate change issues. Gen-

erally, larger cities, with their robust economies and dense populations, tend to excel in climate change initiatives compared to smaller ones [40,41].

With the world increasingly confronting climate change risks, there is a pressing necessity to draw insights from past and current adaptation approaches. This involves comprehending the mechanisms of adaptation and recognizing the constraints faced by different agents involved in these endeavors. In this context, trust has become one of the foremost issues in research about climate change. Research on trust, on its role in identity formation and social group, provides important lessons for climate change research and individuals' adaptive capacity [37,39,42].

Trust has been demonstrated to exhibit a positive association with the adoption of pro-environmental behaviors, encompassing both public and private domains [17], as well as the engagement in collective initiatives addressing climate change [22]. Indeed, trust can function as a heuristic in climate change decision-making, aiding individuals in navigating complex environments [23,24].

Given the expansive magnitude and potentially disastrous repercussions of climate change, recent studies have extensively explored how people's emotional responses to information about climate change impacts and potential solutions influence the decision-making processes pertaining to climate change. Climate change decision-making delineates choices based on the underlying motives or objectives, whether focused on mitigation or adaptation, held by decision-makers [2]. It includes any explicit decisions made by an individual or a collective entity (e.g., individuals, households, communities, organizations, or societies) that hold implications within the context of systems influencing or being influenced by climate change. Consider an individual's decision regarding their transportation mode (such as choosing between a personal vehicle, public transit, or a bicycle): this decision holds relevance to climate considerations due to its impact on the individual's greenhouse gas emissions related to transportation, even if the individual is not consciously contemplating climate change when making the choice.

As already anticipated, climate change decision-making exhibits variations across multiple dimensions, with one of the most apparent distinctions being the locus of decision-making, spanning from individual and household levels to organizational entities [2]. The scope of climate change decision-making extends across diverse domains and sectors, including, but not limited to, transportation, energy (both production and consumption), agriculture, water management, and disaster preparedness [2]. It is noteworthy that climate change decisions exhibit differentiation also concerning the dimensions of time, discerning between the short- versus long-term impacts decisions. The time-related implications of a decision are, to some extent, also contingent upon the frequency with which that decision needs to be routine and repeated versus infrequent [2].

In the realm of climate change decision-making, the literature indicates that trust in governments and political institutions has also been found to positively associated with the uptake of public and private pro-environmental behaviors [17]. For example, Ross et al. [43] tested a model of trust on risk perception and the taking of recycled water. They found that high trust in the water authority is associated with low-risk perception and a high acceptance to recycling water. In contrast, low trust in the water authority was related to high-risk perception and a poor acceptance of climate change-related action. According to Siegrist et al. [44], individuals who lack information about a hazard will evaluate the risk based on their trust in responsible risk managers. Therefore, trust play a vital role in mediating the relationship between experts' risk evaluation and community's risk perception.

Certain forms of adaptation occur as responses by individuals to climate threats, often prompted by specific extreme events. Others are carried out by governments representing society, sometimes in anticipation of change, but frequently in reaction to individual incidents. Consequently, adaptation processes entail the interconnectedness of agents through their reciprocal interactions among themselves, within the institutions they belong to, and with the resources they rely on. The nature of these interactions has been pivotal in

human ecology and geography, microeconomics, as well as anthropological and political sciences. Each discipline has developed theories on trust dynamics in CCA and decision-making related to climate change, but the different emphasis of each discipline has led to a piecemeal view of the importance of trust. For these reasons, we conducted a literature review examining the implications of trust dynamics in climate change decision-making. The aim of this literature review is to analyze the impact of trust on decision-making processes in the context of CCA. On this basis, we endeavor to provide further insights into the role of trust in CCA processes and climate change decision-making.

### 1.2. Research Objectives

We conducted a literature review, examining the implications of trust dynamics in climate change decision-making. The purpose was to analyze the outcomes of pertinent studies, systematize the deriving knowledge, and identify potential gaps in the existing literature, as to identify future directions for improving research and practice on decision-making and the development of trust in in the context of adaptation to climate change.

Specifically, the present literature review was conducted to:

(1)　Analyze how psychological trust dynamics affects the different phases of CCA: preparing the ground for adaptation, assessing climate change risks and vulnerabilities, identifying adaptation options, assessing and selecting adaptation options, implementing adaptation, and monitoring and evaluating adaptation.
(2)　Analyze how psychological trust dynamics are linked to climate change decision-making processes among communities, governments, and institutions.

## 2. Methods

This literature review has been carried out in order to provide a comprehensive understanding of trust dynamics in the realms of CCA. In the preliminary phase, we systematically examined contemporary literature pertaining to the role of trust in decision-making concerning CCA, with the aim of identifying investigations employing rigorous methodologies. Then, we selected the appropriate and relevant search keywords (e.g., trust, community, natural hazard, climate change adaptation, decision-making). This methodological approach sought to enrich and contemporize the knowledge repository within this discipline, thereby securing its pertinence for both present and future cohorts. The selection of keywords was undertaken to encompass diverse dimensions of CCA and trust, and decision-making, spanning across community and institutional spheres. Subsequently, we performed searches in academic databases (PubMed/Medline, ISI Web of Science, and SCOPUS). Empirical studies that investigated the role of trust dynamics in the realms of CCA-related decision-making were deemed as eligible based on the following inclusion criteria: (1) they employed a cohort, case–control, cross-sectional study, and/or experimental design. Publications were excluded if (1) they were not original articles (e.g., proceeding, review, opinion paper, or dissertation), and (2) they did not specifically focus on climate change. After conducting a thorough scrutiny of the retrieved literature, we examined the studies pertaining to the topic. The strength of utilizing this methodology resides in its thorough integration of scientific articles that pertain to the central topic under examination in this literature review. Conversely, its drawback is the omission of grey literature, wherein significant studies might exist that could facilitate a more profound understanding of the phenomenon.

## 3. Results

The subsequent sections will present detailed results of the analysis of the articles on trust, discussed according to the CCA phases of the proposed review framework. Regarding the cycle of the CCA, the analysis reveals that all selected papers focused only on the last two phases of the whole cycle (implementing adaptation, and monitoring and evaluating adaptation). This is understandable as the initial phases of the cycle (preparing the ground for adaptation, assessing climate change risks and vulnerabilities, identifying adaptation

options, and assessing and selecting adaptation options) prioritize only the formulation of strategies for climate change management by regional or local authorities.

During the earlier stages of the CCA cycle, authorities predominantly concentrate on laying the foundation for effective adaptation efforts. This involves a thorough assessment of climate change risks and vulnerabilities, as well as the identification and selection of suitable adaptation options. Moreover, this period is marked by a strategic and proactive examination of the potential consequences of climate change. Despite the equal significance of psychological factors in influencing community responses to climate change, little attention is devoted to these aspects during the initial phases of the CCA cycle. In essence, while the latter stages of the CCA cycle are crucial for the practical implementation and assessment of psychological adaptation measures, it is imperative to recognize the importance of addressing psychological dimensions at earlier stages. Neglecting psychological factors during the initial phases may impede effective community engagement and hinder the development of robust adaptation strategies that resonate with local populations. Therefore, future research and policy efforts should strive to integrate psychological considerations into all stages of the CCA cycle to ensure comprehensive and effective climate change adaptation initiatives.

*3.1. Selected Articles*

This review includes 25 studies reported in peer-reviewed journal articles that have addressed the topic. Of the 25 studies, 19 were quantitative research design, 5 were mixed methods studies, and 1 was an experimental study. Concerning the demographic composition of the included studies, among the 25 scrutinized papers, 17 entail participants from a community of place—a cohort characterized by individuals sharing a common physical or virtual environment as the primary nexus for their affiliation and interaction [45] (e.g., the general population); whereas 8 studies involve individuals from a community of interest—a group of stakeholders who construct a collective identity grounded in shared concerns, objectives, and aspirations [46] (e.g., farmers). All of these studies included both male and female participants, and all were published in English in peer-reviewed journals. They represented research conducted in Australia, Bangladesh, Belgium, Brazil, China, Finland, Iceland, India, Indonesia, Iran, Korea, Malaysia, Nepal, Netherlands, Nigeria, Norway, Spain, Serbia, Sweden, Spain, Swiss, United States, and United Kingdom. Therefore, these papers cover research conducted on a range of countries across North America, Oceania, South Asia, Europe, South America, and Africa in a gradually decreasing way. However, there is a lack of studies exploring trust dynamics in South Africa and in North Asia. Figure 1 shows the graph with the distribution of the publications found on the research topic considering each country. The included studies extend over the period from 2011 to 2023.

Concerning the characteristics of studies referring to climate change decision-making, studies were also classified with reference to: the locus of decision-making (e.g., individual versus household versus organization); domains and sector (e.g., transportation and energy, agriculture, water, and disaster management); scale (e.g., small versus large impact decisions); time (e.g., short- versus long-term impacts); and frequency (e.g., routine and repeated actions versus one-off or infrequent decisions) [2]. Regarding the locus of decision-making, out of the 25 papers analyzed, 16 of them had the locus on individual, followed by household (7), organization (2), and one study had the locus on both individual and organization. Concerning the domains and sector, 9 focused on disaster management, followed by energy sector (7), agriculture sector (6), and for the domains of transportation and water there were only two study, as one for mixed domains. Regarding the dimension of scale, almost all of the studies (14) assessed large impact decisions, whereas only nine studies assessed behaviors characterized by small impact decisions. With respect to the time, almost all of the studies (21) assessed behaviors with long-term impacts, whereas only two studies assessed behaviors with short-term impacts. Regarding the dimension of frequency, almost all of the studies (21) assessed behaviors with routine and repeated

actions, whereas only two studies assessed behaviors with infrequent decisions. Two studies could not be categorized based on the dimensions of scale, time, and frequency.

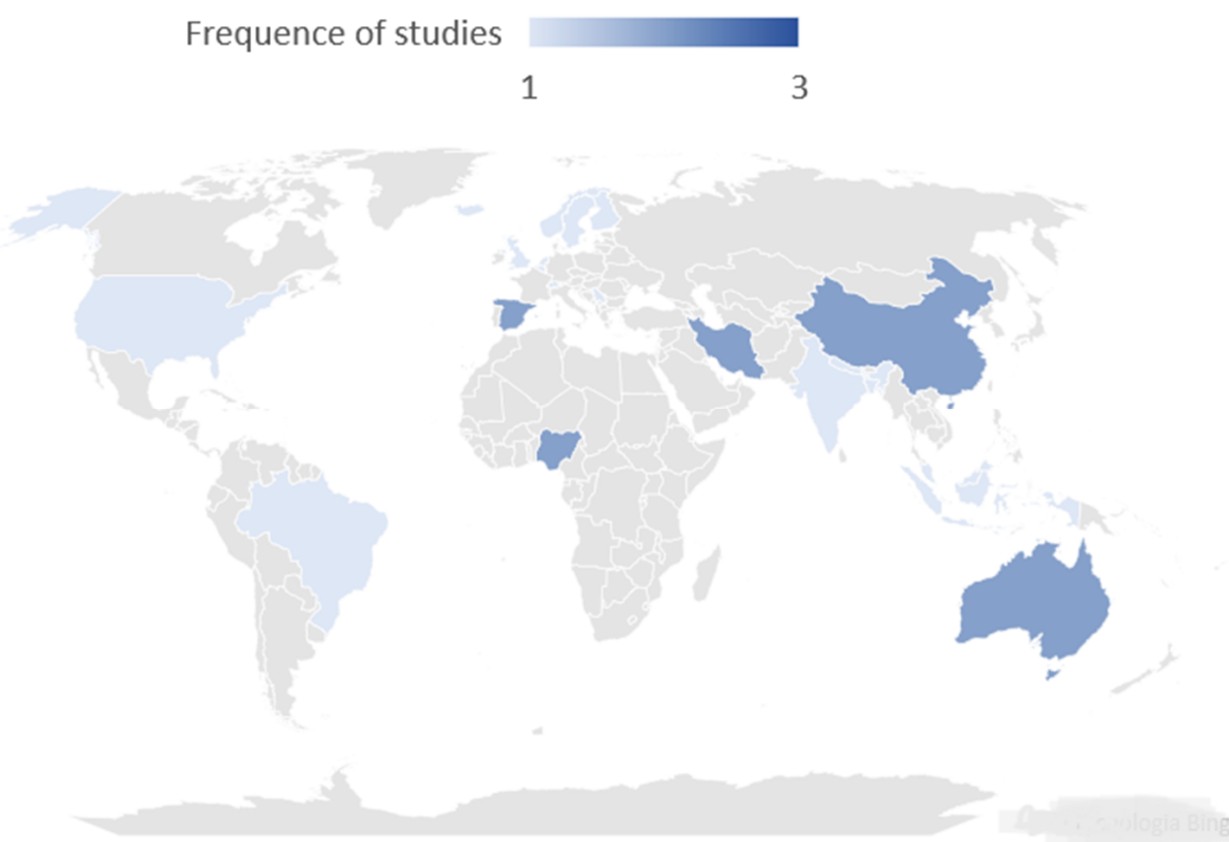

**Figure 1.** Distribution of the publications based on country.

*3.2. Trust in the Implementing Adaptation Phase*

In the implementing adaptation phase of CCA, adaptation policies should be implemented through a strategy planned by regional or local authorities. In this phase, the implementation of favored adaptation options serves to assist regions or local authorities in managing their adaptation strategy and action plan to counter climate change. At this phase, trust can assume a key role in CCA, and its function may depend on various aspects, and it can influence different types of decisions (such as the community's willingness to adopt climate change environmental solutions planned by authorities; [37]).

Most of the studies on trust in the implementing adaptation phase used a quantitative research design (14), followed by a mixed methods research design (3), and by one experimental study.

Quantitative studies underline that trust influences the willingness to adopt renewable energy technologies by individuals, households, and enterprises. Specifically, Akinwale and Adepoju [37] showed that trust in institutions influences the willingness to adopt climate change environmental solutions by micro- and small-enterprises. In addition, Fairbrother et al. [39] highlighted that Europeans who exhibit high levels of trust in institutions are considerably more inclined to endorse fossil fuel taxes, particularly if they also acknowledge the reality and risks associated with human-induced climate change. Also, Fairbrother et al. [47] found that individuals with higher institutional trust were significantly less skeptical of the benefits of tax policies on CCA. Concerning community trust, Akter [38] emphasized how individuals' community trust is linked to the willingness to pay for cyclone risk reduction. In addition, Saptutyningsih et al. [48] found that community trust is closely related to farmers' willingness to contribute financially to the CCA adaptation process. Finally, Wang et al. [49] found notable variations in the adoption of climate change

adaptation measures among farmers with varying degrees of trust, encompassing both community and institutional trust. Notably, local farmers demonstrating higher levels of trust were inclined to execute a greater number of climate change adaptation strategies.

Other quantitative studies [50,51] underlined that both community and institutional trust are precursors of risk perception, which, in turn, is a precursor of mitigation behaviors in the field of CCA. For example, Vainio and Paloniemi [52] found that institutional trust is a precursor of climate-friendly actions. They also emphasized that lack of trust in the existing political system motivates individuals to engage in non-institutional actions in the realm of climate change initiatives. Also, De Vocht et al. [53] emphasize the significant influence of trust on risk perception, particularly in situations where knowledge about the risk is limited.

Referring to the scientific-technological trust, Cologna et al. [54] found that trust in climate scientists predicted willingness to engage in both low- and high-impact CCA behaviors. Also, Kettle and Dow [21] have emphasized that trust in scientists as significant figures influences a community's CCA strategies. In addition, Park [55] showed the significant role of trust in energy technologies in regard to the social acceptance of the diffusion, distribution, and the success of renewable energy technologies. Finally, Rahmani and Bonyadi Naeini [56] found that trust in technology is a precursor of the solar energy technologies acceptance.

Mixed methods studies underline that community trust significantly and positively influences adaptation actions [57]. This implies that residents who possess a trusting and supportive network, along with a sense of efficacy in effecting communal change, are inclined to undertake proactive measures in the realm of CCA. The study conducted by Le et al. [42] elucidates the significance of trust in scientists as a pivotal determinant influencing communities' positive disposition and endorsement of CCA strategies, particularly in the context of addressing acute crises and restoring coral reef ecosystems. This is particularly pronounced when participants possess restricted knowledge and comprehension of coral restoration. So, authors suggest that trust is an important determinant of public acceptance and an additional factor to strengthen public support. Nikolakis and Guðjónsson [58] also showed that full communication on CCA themes builds and maintains institutional trust, and that institutional trust enables the achievement of goals for climate-related action. Finally, an experimental study [59] showed the importance of institutional trust in the low-carbon technologies project acceptance for the inhabitants of Netherlands and United Kingdom.

In summary, trust dynamics in the implementing adaptation phase are related to individuals' proactive behaviors within the field of CCA. This happens especially when people have an understanding that a behavior change is for public motives and not for someone else's advantage [50,60]. This implies also that the impact of trust on CCA policy attitudes is manifested through its influence on perceptions of policy honesty. Overall, these findings support the interconnectedness of institutional trust, community trust, scientific-technological trust, CCA policy support, and climate change decision-making. Table 1 summarizes the main findings on the role of trust in the implementing adaptation phase of CCA.

Table 1. Themes related to trust in the implementing adaptation phase.

| Studies | Type of Study | Country | Type of Decision-Making | Locus of Decision-Making | Domains and Sector | Scale | Time | Dimension of Frequency | Type of Trust | Themes Identified on Trust in CCA |
|---|---|---|---|---|---|---|---|---|---|---|
| Akinwale and Adepoju, 2019 [37] | Quantitative | Nigeria | Adopt renewable energy technologies | Organization | Energy | Large | Long term | Routine | Institutional | Trust influenced the willingness to adopt renewable energy technologies for micro and small enterprises. |
| Akter, 2020 [38] | Quantitative | Bangladesh | Pay for polder improvement. | Household | Disaster management | Small | Long term | Routine | Community | The readiness to invest in cyclone risk mitigation showed notable fluctuations based on the levels of trust assessed at the personal level. Elevated levels of trust were associated with a greater inclination to pay for risk reduction measures. |
| Choon et al., 2019 [50] | Quantitative | Malaysia | Adopt mitigation and adaptation initiatives. | Individuals | Energy | Small | Short term | Routine | Institutional Community | Increased social trust will resulted in heightened risk perception, which significantly shaped public responses to, addresses, and supports climate change mitigation and adaptation efforts. |
| Cologna et al., 2022 [54] | Quantitative | Swiss | Adopt pro-environmental behaviors. | Individuals | Energy | Large | Long term | Routine | Scientific-technological | Higher levels of trust in climate scientists predicted the willingness to engage in both low- and high-impact mitigation behaviors. |
| De Vocht et al., 2015 [53] | Quantitative | Norway, Spain, Serbia, Belgium | Behavioral intentions of CCA strategies. | Individuals | Disaster management | Small | Short term | Routine | Institutional | Trust in the government influenced climate change related behavioral intentions. |
| Fairbrother et al., 2021 [39] | Quantitative | Sweden, Spain, South Korea, China | Protect the environment (e.g., tax policies). | Household | Disaster management | Large | Long term | Routine | Institutional | People with high political trust were significantly less skeptical of the benefits of tax policies related to climate change. |
| Fairbrother et al., 2019 [47] | Quantitative | 23 European countries | Environmental attitudes (e.g., carbon tax). | Household | Energy | Large | Long term | Routine | Institutional | People who lived in countries with high political trust tended to be much more supportive of fossil fuel taxes. |

**Table 1.** *Cont.*

| Studies | Type of Study | Country | Type of Decision-Making | Locus of Decision-Making | Domains and Sector | Scale | Time | Dimension of Frequency | Type of Trust | Themes Identified on Trust in CCA |
|---|---|---|---|---|---|---|---|---|---|---|
| Haas et al., 2021 [57] | Mixed | India | Actions to deal with flooding and water scarcity. | Household | Water | Large | Long term | Routine | Community | Trust significantly and positively influenced adaptation actions. |
| Jin, 2023 [51] | Quantitative | Korea | Collaborative behavior to address climate change. | Individuals | Disaster management | / | / | / | Community | Trust moderated the causal relationship between societal risk perception and climate change related behavioral intentions. |
| Kettle and Dow, 2016 [21] | Quantitative | United States | Support for the development of plans. | Individual/ Organization | Disaster management | / | / | / | Scientific-technological | Trust played a pivotal role in shaping individuals' endorsement of climate change adaptation measures. |
| Le et al., 2022 [42] | Mixed | Australia | Accept coral restoration projects. | Individuals | Disaster management | Large | Long term | Routine | Institutional | Trust emerged as the predominant factor influencing public support for coral restoration endeavors. |
| Nikolakis and Guðjónsson, 2021 [58] | mixed methods research design | Iceland | Voluntary and inter-organizational climate cooperation. | Organization | Disaster management | Large | Long term | Routine | Institutional | Trust emerged as a crucial element fostering collaboration in endeavors related to climate action. |
| Park, 2021 [55] | Quantitative | Korea | Perspectives toward particular energy technologies. | Individuals | Energy | Large | Long term | Routine | Scientific-technological | Trust was a key determinant of the public's desire to adopt renewable energy technologies. |
| Rahmani and Bonyadi Naeini, 2023 [56] | Quantitative | Iran | Applying solar energy technologies. | Individuals | Agriculture | Small | Long term | Routine | Scientific-technological | Trust had a positive effect on solar energy technologies usage intention in agriculture industry. |
| Saptutyningsih et al., 2020 [48] | Quantitative | Indonesia | Adaptation strategies for climate change adaptation. | Household | Agriculture | Small | Long term | Routine | Community | High level of trust was positively correlated with the willing to contribute financially to the adaptation process. |
| ter Mors and van Leeuwen, 2023 [59] | Experimental | Netherlands, United Kingdom | Acceptance of low-carbon technologies. | Individuals | Transportation | Large | Long term | Routine | Institutional | Trust was linked to acceptance of the low-carbon technology project. |

**Table 1.** *Cont.*

| Studies | Type of Study | Country | Type of Decision-Making | Locus of Decision-Making | Domains and Sector | Scale | Time | Dimension of Frequency | Type of Trust | Themes Identified on Trust in CCA |
|---|---|---|---|---|---|---|---|---|---|---|
| Vainio and Paloniemi, 2013 [52] | Quantitative | Finland | Climate-friendly action. | Individuals | Mixed | Large | Long term | Routine | Institutional | The belief in climate change mediated the effect of post-material values, trust and knowledge on climate-friendly action. |
| Wang et al., 2021 [49] | Quantitative | China | Climate change adaptation strategies. | Household | Agriculture | Large | Long term | Routine | Institutional Community | The levels of trust individuals had in both interpersonal relationships and institutions strongly influence the selection of climate change adaptation strategies among farmers. |

### 3.3. Trust in the Monitoring and Evaluating Adaptation Phase

The monitoring and evaluating adaptation phase of CCA occurs after the implementation process. This phase assists in assessing the effectiveness of adaptation measures and identifying any unforeseen side effects that may have occurred. The efforts of regional or local authorities are directed towards undertaking monitoring and evaluation activities, as it serves both learning and accountability. Monitoring helps authorities in assessing the efficacy of adaptation measures and identifying any unforeseen adverse consequences they may have been engendered. Also, in this phase, trust can assume a key role in promoting the implementation and maintenance of the adaptation measures.

Most of the studies on trust in the monitoring and evaluating adaptation phase used a quantitative research design (5), followed by mixed methods research design (2).

Quantitative studies underline that both community and institutional trust were predictors of CCA-related behaviors. The study by Azadi et al. [61] showed that both these types of trust can influence beliefs and risk perception [61], as trust facilitates the recognition and understanding of climate change. Also, Budhathoki et al. [62] found that individuals who have a strong sense of trust in government-led adaptation initiatives perceive lower levels of risk compared to those with less trust. Furthermore, their findings indicate a positive correlation between institutional trust and farmers' climate-related hazards prevention behavior. However, in certain instances, despite farmers expressing confidence in governmental adaptation plans and policies, they remained inclined to implement their own adaptation measures.

Bakaki and Bernauer [27] showed that low levels of trust in public institutions have a strong negative impact on the public's willingness to pay for forest conservation. Along these lines, the willingness to pay increases with the trust in government. Also, Berry et al. [63] found that trust is predictive of CCA strategies (e.g., create plans for managing risks associated with natural disasters; aim to adopt technologies to diminish emissions; aspire to generate environmentally friendly energy; or express enthusiasm for installing wind turbines on their property for energy generation).

Devine-Wright and Batel [64] outlined a profile of those who do not have a social acceptance of energy infrastructure. Their research revealed that individuals with the least trust in grid operators, who manage the transportation of electricity from production to consumption, also exhibited the lowest inclination to act regarding personal demand or nearby power line proposals. Moreover, they displayed the least support for European grid operators and expressed minimal concern about climate change. These individuals were more commonly younger, and/or less inclined to vote.

Mixed methods studies emphasize that countries with elevated levels of both institutional and community trust exhibit greater apprehension regarding the issue of global warming compared to countries with lower trust levels. Notably, individuals residing in low-income countries with heightened social trust demonstrate a greater degree of concern about global warming than respondents in high-income nations [65]. Ultimately, the disparities within countries between groups characterized by low and high levels of both community and institutional trust are substantially more pronounced in high-income nations compared to low-income ones [65]. Furthermore, Ekoh et al. [66] have emphasized the importance of a lack of trust in the mobility adaptation strategies for residents impacted by extreme events. Individuals who have a low trust in institutions would prefer that migrants facilitate movement of migration and relocation by themselves. Therefore, it is essential for governmental agencies to prioritize the establishment of trust before embarking on government-led relocation initiatives.

In summary, we found that also in the monitoring and evaluating adaptation phase, trust is related to proactive behaviors within the field of CCA. From these findings, we deduce that public trust holds greater significance in the formulation of climate change decision-making strategies. Also, trust is positively associated with adaptation behaviors in response to natural hazards. Table 2 summarizes the main findings on the role of trust in the monitoring and evaluating adaptation phase of CCA.

**Table 2.** Themes related to trust, identified during the monitoring and evaluating adaptation phase.

| Studies | Type of Study | Country | Type of Decision-Making | Locus of Decision-Making | Domains and Sector | Scale | Time | Dimension of Frequency | Type of Trust | Themes Identified on Trust in CCA |
|---|---|---|---|---|---|---|---|---|---|---|
| Azadi et al., 2019 [61] | Quantitative | Iran | Shifting planting dates; diversifying into other crops; use of manure and compost; use of crop rotation; changing timing of chemical inputs; change the amount of chemical inputs. | Individuals | Agriculture | Small | Long term | Routine | Institutional | Trust was effective in driving farmers' climate adaptation behaviors. |
| Bakaki and Bernauer, 2016 [27] | Quantitative | Brazil | Pay for forest conservation. | Individuals | Disaster management | Large | Long term | Infrequent | Institutional | Limited trust in public institutions significantly diminished the public's readiness to contribute financially to forest conservation efforts. |
| Berry et al., 2011 [63] | Quantitative | Australia | Adaptation through planning and managing property; intention to adapt practices; desire to produce green power. | Individuals | Agriculture | Small | Long term | Routine | Institutional Community | Higher levels of trust were linked to adaptive practices for climate change. |
| Budhathoki et al., 2020 [62] | Quantitative | Nepal | Changes to planting dates and crop varieties and increasing the use of fertilizers and pesticides. | Individuals | Agriculture | Small | Long term | Routine | Institutional | Trust influences flood risk perception. Risk perception, in turn, mediated the relationship between trust and farmers' intended flood adaptation strategies. |
| Devine-Wright and Batel, 2017 [64] | Quantitative | England | Acceptance of low carbon infrastructure. | Individuals | Energy | Large | Long term | Routine | Institutional Community | Individuals who exhibited high trust with respect to local, national, and global levels demonstrated the highest willingness to decrease energy consumption. Conversely, those with weaker connections were least likely to trust grid companies. |
| Ekoh et al., 2023 [66] | Mixed | Nigeria | Migration and relocation intention. | Individuals | Disaster management | Small | Long term | Infrequent | Institutional | High levels of trust were linked to government-aided relocation strategies. |
| Imbulana Arachchi and Managi, 2022 [65] | Mixed (indirect and quanti-tative) | 30 developing and developed countries | Reduce carbon dioxide emissions. | Household | Energy | Large | Long term | Routine | Institutional Community | Social trust was associated with higher concern about the global warming issue. |

## 4. Discussion

This review has analyzed articles focused on trust dynamics during the CCA cycle to address the existing multiple research gaps to better understand the role of trust in the different phases of CCA and how trust dynamics are linked to individuals' climate change decision-making.

Despite challenges posed by variations in the definition and assessment of trust, drawing definitive conclusions from the body of research can be challenging. Nonetheless, the majority of studies facilitated the addressing of research objectives and delineated positive associations between trust and fundamental constituents of decision-making pertaining to climate change. Results of this review, thus suggest that community trust, institutional trust, and scientific-technological trust serve as essential components facilitating communities in CCA and climate change decision-making.

Regarding the cycle of the CCA, the analysis reveals that all selected articles focused on the two last stages (namely, implementing adaptation, and monitoring and evaluating adaptation) of the whole cycle. Notably, in the early phases of CCA, authorities focus on pinpointing optimal solutions that contribute to climate risk assessment and conduct a strategic and proactive analysis of the repercussions of climate change. Despite their equal importance, minimal attention is given to the psychological factors influencing the community at this stage.

Specifically, it emerges in papers focusing on the stage of implementing adaptation, it emerges that trust influences multiple adaptive actions. Precisely, it impacts the propensity to adopt renewable energy technologies [37], the inclination to invest in cyclone risk mitigation [38], the readiness to participate in both low- and high-impact mitigation efforts [54], and the readiness to support fossil fuel taxation [39]. Moreover, trust also contribute to heightened risk perception, exerting a substantial influence on public responses to, efforts to address, and backing of initiatives for climate change mitigation and adaptation [50]. Furthermore, trust emerged as one of the most relevant factors influencing the public acceptance (i.e., attitudinal engagement) of CCA measures [42]. In papers focusing on the stage of monitoring and evaluating adaptation, it emerges that trust plays a pivotal role in motivating CCA behaviors [61]. More precisely, increased levels of trust are associated with the adoption of adaptive measures for climate change [63] as well as with government-supported relocation strategies [66].

While trust holds varied implications across the last stages of the CCA cycle, a thorough comprehension of research findings is imperative to interpret these results. Indeed, upon comprehensive examination of study outcomes, a recurring theme becomes apparent: individuals retain the potential to nurture trust throughout the entire CCA cycle, with the cultivation of trust substantially augmenting capacity for climate change mitigation. Hence, it is essential for institutions to acknowledge that cultivating trust is imperative throughout all phases of the cycle to ensure the continuous facilitation of this pivotal dynamic, thereby benefiting CCA.

Given the immense scale and potentially catastrophic repercussions of climate change, recent studies have thoroughly investigated how individuals' emotional reactions to information about its impacts and potential remedies affect decision-making concerning this global challenge [67,68]. This review of the literature unveiled that trust is also a key component for climate change decision-making, fostering decisions with immediate and direct implications for climate systems. Trust indeed represents both an objective and a means for the development of community decision-making. The literature review revealed that all the domains of trust (institutional, community, and scientific-technological) may play a crucial role in decision-making within the realm of CCA. The analysis reveals that trust dynamics influence different adaptive actions linked to renewable energy technologies adoption, low- and high-impact mitigation behaviors, and paying taxes [37,39,54]. In addition, trust dynamics contributes to heightened risk perception, which substantially shapes public responses in support of climate change mitigation and adaptation initiatives [42,50].

Accordingly, Moreno et al. [69] argued that for communities to exhibit resilience in the face of natural hazards, a strong foundation of trust is needed. As observed elsewhere [1], trust and the perception of trustworthiness in the context of disaster risk reduction and CCA are dependent upon the responsiveness of institutions and community members, based on their capacity to communicate openly and handle the situation. Within this context, Kitagawa [70] examined the significance of promoting decision-making and participation through collaborative projects in community learning for disaster preparedness, emphasizing the need for the population's commitment and cooperation to prevent and mitigate anticipated large-scale disasters. Given that human decision-making regarding climate change plays a pivotal role in shaping both individual and societal responses to environmental risks [71], these partnerships foster empowering and reliable connections between community members and authorities. Through collaborative efforts, both parties can familiarize themselves, exchange ideas and information, and jointly develop preparedness plans. At the community level, developing these types of shared and participatory approaches to global issues such as climate change is crucial for building trust, raising awareness, and enhancing the general public's knowledge base. Hence, in order to foster a positive and virtuous culture of trust in the field of climate change, it is imperative to take into account all three domains of the trust dynamics: effective communication that aligns with the context and participants, active engagement of community members as catalysts for change, and the establishment of adequate feedback mechanisms to collectively assess the impact of participatory processes. With respect to this, this review yields several practical implications. Its results emphasize the importance of improving risk communication between governmental bodies and the public. Specifically, local authorities and experts are required to facilitate the distribution of risk-related information and enhance the safety standards of a region through a participatory, bottom-up approach. However, effective communication concerning climate change entails more than merely informing citizens; it requires an assessment of their understanding of the messages conveyed. Those engaged in communication must be aware of the community's needs, preferences, the level of comprehension, and favored communication channels. Highlighting the capability, willingness, and the entitlement of individuals to actively engage in CCA demonstrates how professionals can benefit from their involvement. Additionally, given current climate forecasts, community participation in practical training such as civil protection exercises becomes increasingly essential to improve the prevention, preparedness, and response to natural disasters. Moreover, comprehending individuals' perceptions of risks related to climate change is critical for effectively disseminating information. The ultimate goal is to cultivate trust in authorities and enhance the ability to respond to significantly impactful changes in climate, thereby strengthening community resilience. Identifying the needs of citizens constitutes the initial and indispensable step toward promoting the exchange of intervention policies aimed at fortifying resilience and CCA.

This review thus represents a first attempt to organize knowledge on trust dynamics in the domain of CCA-decision making. Preceding this review, the implications stemming from psychological trust dynamics had not undergone thorough scrutiny throughout the entirety of the CCA cycle, leading to a deficit in the integration of studies within this framework. Also, the review stands as a pivotal point of reference for scholars, prompting the initiating of new investigations into all facets of the CCA cycle, where the dynamics of psychological trust have yet to be thoroughly investigated.

Based on the review findings, it is essential for stakeholders involved in CCA to cultivate and advocate for a "culture of trust" within communities. Establishing a culture of trust faces challenges due to the intricate nature of the concept and the necessary conditions for its realization. However, as with many complex issues, transparent and open communication emerges as a powerful tool that is accessible to all communities. Truthful communication regarding the need for and the implementation of CCA measures can address citizens' concerns and mitigate inappropriate responses when implementing strategies for managing extreme events and adapting to climate change. Cultivating this

culture of trust is vital for enhancing community resilience and fostering a cohesive bond where all members collectively strive to ensure everyone's safety.

## 5. Conclusions

The present review facilitated the addressing of research objectives and delineated positive associations between trust and fundamental constituents of decision-making pertaining to climate change. Results of the review provides evidence concerning the effects of trust dynamics on CCA and on individuals' climate change decision-making: institutional, community, and scientifical-technological trust are associated to climate change decision-making in the CCA cycle. As a result, this review constitutes an initial effort to consolidate and systematize the existing evidence pertaining to the influence of trust on CCA. The results suggest that trust assumes a crucial role in the latter stages of CCA, exerting an impact on the effectiveness of government-driven implementation, monitoring, and evaluation of adaptation strategies in response to climate change. Consequently, there is a pressing need to foster a "culture of trust" at both the community and institutional levels, to develop effective adaptation strategies and positively shape decision-making processes related to climate change.

## 6. Limitations

This critical review has some limitations to be addressed. Firstly, the evaluation encompasses trust across various phases of CCA as distinct entities. This approach is dictated by data limitations, such as the absence of longitudinal studies, but it might obscure the complex inter-relationships and interdependencies among all the phases. Secondly, the identified studies included evaluation of trust only in the two last stages of CCA. Consequently, there is a need to develop further study to enrich the understanding of the role of trust in CCA-related decision making in the other phases. Thirdly, studies were conducted in some countries, but they did not cover the entire globe; so, it was not possible to assess how trust dynamics impact on climate change decision-making in a global way. Fourthly, there is a paucity of research dedicated to scientific-technological confidence. Hence, it is imperative to promote the dissemination of novel studies addressing this subject with the objective of enhancing comprehension regarding the significance of scientific-technological trust in societal adaptation to climate change. In conclusion, the review consolidates studies utilizing diverse methodologies to evaluate trust and its associated factors in CCA, encompassing both ad hoc and validated instruments. This diversity in assessment methods limits the potential for direct comparison of study results. This limitation notwithstanding, the review distinctly emphasizes the interconnectedness of institutional, community, and scientific-technological trust with the efficacy of climate change adaptation initiatives.

## 7. Future Directions

The absence of longitudinal and experimental evidence illustrating causal relationships between trust dynamics and individuals' decision-making in CCA should not be construed as evidence of non-existence. It is imperative to conduct additional research to elucidate the causal connections between trust and the efficacy of CCA initiatives. This endeavor will likely demand the application of innovative research methodologies, such as longitudinal and experimental designs, capable of exploring the specific factors influencing trust within the realm of CCA.

**Author Contributions:** Conceptualization, R.C.B. and A.S.; methodology, R.C.B.; formal analysis, R.C.B. and A.S.; data curation, R.C.B., A.S. and S.R.; writing —original draft preparation, R.C.B. and A.S.; writing—review and editing, A.S. and S.R.; supervision, A.S.; project administration, A.S. All authors have read and agreed to the published version of the manuscript.

**Funding:** This study was carried out within the RETURN Extended Partnership and received funding from the European Union—NextGenerationEU (National Recovery and Resilience Plan—NRRP, Mission 4, Component 2, Investment 1.3—D.D. 1243 2/8/2022, PE0000005).

**Informed Consent Statement:** Not applicable.

**Data Availability Statement:** Not applicable.

**Conflicts of Interest:** The authors declare no conflicts of interest.

## Abbreviations

| CCA | climate change adaptation |
| UNDP | United Nations Development Program |

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
