# Peer review of "Psychological Trust Dynamics in Climate Change Adaptation Decision-Making Processes: A Literature Review"

_sustainability, doi:10.3390/su16103984_

Round 1
Reviewer 1 Report
Comments and Suggestions for Authors
This manuscript represents a valuable contribution to the study of the role of trust in climate change adaptation processes. The authors employ a comprehensive approach to analyzing literature on various aspects of trust, including its psychological definition, social interactions, and institutional frameworks. The work is distinguished by a deep understanding of interdisciplinary connections and the significance of trust as a foundational element in relationships between individuals, groups, and institutions. However, it would be extremely important to pay special attention to issues of public and government trust and their connection to "scientific-technological trust".
Comments to the manuscript:
1. In the "Keywords" section, the repetition should be eliminated by keeping either "decision-making" or "climate change decision-making". If the first option is more general and covers a wide range of decisions, the second is more specific and focused on decisions related to climate change. I recommend choosing the term that best matches the main theme of the research.
2. The absence of section "1.2." should be corrected by adding the necessary section or renumbering the existing ones to avoid confusion among readers.
3. In point "1.3. Research Objectives", it would be sensible to reformulate the goals and replacing it with a more specific description of the research intentions. For example: "The aim of this study is to analyze the impact of trust on decision-making processes in the context of adaptation to climate change." Why make “understanding” the issue your goal?
4. In section "2. Methods", it is necessary to expand the description of the methodology by specifying the scope and time frame of the literature analyzed, as well as the principles for classifying the sources used. It should also be indicated whether the methods applied are standard for this field of research or were developed specifically for this study by the authors.
5. In section "3. Results", the phrase "In the following sections" should be specified, for example: "The subsequent subsections will present detailed results of the analysis." It is also important to classify the data presented in tables more clearly and ensure their direct connection with the text of the research.
6. The text indeed should pay more attention to describing types of trust, especially scientific-technological trust, as it plays an important role in decision-making for adaptation to climate issues. It is necessary to thoroughly analyze how trust in scientific data and technological innovations affects public perception of climate change and readiness for action.
7. In conclusion, it should be clarified what is meant by "scientific-technological trust" and emphasize its importance for timely support of science in the context of climate change.
8. Therefore, in section "6. Limitations and future directions", doubts are expressed as it is necessary to understand the trust between "Scientific-technological trust" and society.
Comments on the Quality of English LanguageLong sentences. Perhaps it needs to be simplified.
Author Response
Response to Reviewer 1 Comments
This manuscript represents a valuable contribution to the study of the role of trust in climate change adaptation processes. The authors employ a comprehensive approach to analyzing literature on various aspects of trust, including its psychological definition, social interactions, and institutional frameworks. The work is distinguished by a deep understanding of interdisciplinary connections and the significance of trust as a foundational element in relationships between individuals, groups, and institutions. However, it would be extremely important to pay special attention to issues of public and government trust and their connection to "scientific-technological trust".
Comments to the manuscript:
Point 1: In the "Keywords" section, the repetition should be eliminated by keeping either "decision-making" or "climate change decision-making". If the first option is more general and covers a wide range of decisions, the second is more specific and focused on decisions related to climate change. I recommend choosing the term that best matches the main theme of the research.
Response: Thank you for the suggestion. We have modified keywords using only terms that best matches the main theme of the research (“trust; climate change adaptation; climate-related hazards; climate change decision-making”).
Point 2: The absence of section "1.2." should be corrected by adding the necessary section or renumbering the existing ones to avoid confusion among readers.
Response: Thank you for the suggestion. We have corrected the error (RL 330).
Point 3: In point "1.3. Research Objectives", it would be sensible to reformulate the goals and replacing it with a more specific description of the research intentions. For example: "The aim of this study is to analyze the impact of trust on decision-making processes in the context of adaptation to climate change." Why make “understanding” the issue your goal?
Response: Thank you for the suggestion. We have modified the research objectives, as follows (RL 331-342):
“We conducted a literature review examining the implications of trust dynamics in climate change decision-making. The purpose was to analyze the outcomes of pertinent studies, systematize the deriving knowledge, and identify potential gaps in the existing literature, so to identify future directions for improving research and practice on decision making and development of trust in in the context of adaptation to climate change.
Specifically, the present literature review was conducted to:
1) Analyze how psychological trust dynamics affects the different phases of CCA: preparing the ground for adaptation, assessing climate change risks and vulnerabilities, identifying adaptation options, assessing and selecting adaptation options, implementing adaptation, and monitoring and evaluating adaptation.
2) Analyze how psychological trust dynamics are linked to climate change decision-making processes among communities, governments and institutions”
Point 4: In section "2. Methods", it is necessary to expand the description of the methodology by specifying the scope and time frame of the literature analyzed, as well as the principles for classifying the sources used. It should also be indicated whether the methods applied are standard for this field of research or were developed specifically for this study by the authors.
Response: Thank you for the suggestion. In the revised version of the manuscript, we have better clarified the methodology behind the review, as follows (RL 346-365):
“In the preliminary phase, we systematically examined contemporary literature pertaining to the subject matter with the aim of identifying investigations employing rigorous methodologies. Then, we selected the appropriate and relevant search keywords (e.g., trust, community, natural hazard, climate change adaptation, decision-making). This methodological approach sought to enrich and contemporize the knowledge repository within this discipline, thereby securing its pertinence for both present and future cohorts. The selection of keywords was undertaken to encompass diverse dimensions of CCA and trust, spanning across community and institutional spheres. Subsequently, we performed searches in academic databases (PubMed/Medline, ISI Web of Science, and SCOPUS). Empirical studies that investigated the role of trust dynamics in the realms of CCA were deemed eligible based on the following inclusion criteria: (1) they employed cohort, case-control, cross-sectional study, and/or experimental design. Publications were excluded if (1) they were not original articles (e.g., proceeding, review, opinion paper, or dissertation), and (2) they did not specifically focus on natural disaster. After conducting a thorough scrutiny of the retrieved literature, we examined the studies pertaining to the topic. The strength of utilizing this methodology resides in its thorough integration of scientific articles that pertain to the central topic under examination in this literature review. Conversely, its drawback is the omission of grey literature, wherein significant studies might exist that could facilitate a more profound understanding of the phenomenon.”
Point 5: In section "3. Results", the phrase "In the following sections" should be specified, for example: "The subsequent subsections will present detailed results of the analysis." It is also important to classify the data presented in tables more clearly and ensure their direct connection with the text of the research.
Response: Thank you for the suggestion. We have modified the Results section.
Point 6: The text indeed should pay more attention to describing types of trust, especially scientific-technological trust, as it plays an important role in decision-making for adaptation to climate issues. It is necessary to thoroughly analyze how trust in scientific data and technological innovations affects public perception of climate change and readiness for action.
Response: Thank you for the suggestion. In the revised version of the manuscript, we have better clarified the role of scientific-technological trust, as follows (RL 137-245):
“According to the findings of Cologna and Siegrist17, there exists a robust correlation between trust in scientists and the engagement in climate-friendly behaviors. This observation is further supported by Tranter and Lester28, who determined that individuals who exhibit trust in scientists as purveyors of environmental information tend to prioritize initiatives addressing climate change. Moreover, research by Leiserowitz and colleagues29 and Tranter and Lester28 underscore that heightened trust in climate scientists and the scientific enterprise as a whole corresponds to an increased likelihood of acknowledging human-induced factors in climate change”
Point 7: In conclusion, it should be clarified what is meant by "scientific-technological trust" and emphasize its importance for timely support of science in the context of climate change.
Response: Thank you for the suggestion. In the revised version of the manuscript, we have better clarified what is meant by "scientific-technological trust", as follows (RL 105-108; 772-777):
“It constitutes a variant of social or institutional trust, characterized by the attribution of impersonal trust to individuals engaged in scientific pursuits and technological enterprises within institutional contexts (such as scientists or technological experts)18.”
…
“Fourthly, there is a paucity of research dedicated to scientific-technological confidence. Hence, it is imperative to promote the dissemination of novel studies addressing this subject with the objective of enhancing comprehension regarding the significance of scientific-technological trust in societal adaptation to climate change.”
Point 8: Therefore, in section "6. Limitations and future directions", doubts are expressed as it is necessary to understand the trust between "Scientific-technological trust" and society.
Response: Thank you for the suggestion. We have modified the Limitations and future directions section.
Reviewer 2 Report
Comments and Suggestions for Authors
As some climate-related risks are not currently directly experienced by people, trust in experts is required if individuals are to acknowledge and evaluate those risks. However, even when the risks of climate change are recognised, choosing between different mitigation and adaptation behaviours can be a complex cognitive task, especially when knowledge of climate science is low and uncertainties are high. Trusting experts can help to alleviate the cognitive complexity of evaluating such risks and related behavioural decisions by serving as a heuristic in decision making. Gaining a deeper understanding of how trust in certain actors influences individuals' mitigation and adaptation behaviours is, therefore, key. As such, this study aims to examine the research related the implications of the trust dynamics in climate change adaptation from different levels of analysis, looking at the different phases of adaptation and considering its importance in climate change decision-making. The results show that trust dynamics influence different communities' adaptive actions linked to renewable energy technologies adoption, low- and high-impact mitigation behaviors, and paying taxes for actions that mitigate climate change.
I think this study is a progress by focusing on the role of trust for climate change mitigation behaviour. Generally, some revision suggestions are listed below:
1) It's better to discuss the role of trust in institutions, scientists, industry, environmental groups and people in different countries, e.g. , developing countries and developed countries.
2) Takeaway for Practice is also encouraged to be included in this paper. It is better to be clear enough to present your policy recommendations for local practice.
Author Response
Response to Reviewer 2 Comments
As some climate-related risks are not currently directly experienced by people, trust in experts is required if individuals are to acknowledge and evaluate those risks. However, even when the risks of climate change are recognised, choosing between different mitigation and adaptation behaviours can be a complex cognitive task, especially when knowledge of climate science is low and uncertainties are high. Trusting experts can help to alleviate the cognitive complexity of evaluating such risks and related behavioural decisions by serving as a heuristic in decision making. Gaining a deeper understanding of how trust in certain actors influences individuals' mitigation and adaptation behaviours is, therefore, key. As such, this study aims to examine the research related the implications of the trust dynamics in climate change adaptation from different levels of analysis, looking at the different phases of adaptation and considering its importance in climate change decision-making. The results show that trust dynamics influence different communities' adaptive actions linked to renewable energy technologies adoption, low- and high-impact mitigation behaviors, and paying taxes for actions that mitigate climate change.
I think this study is a progress by focusing on the role of trust for climate change mitigation behaviour. Generally, some revision suggestions are listed below:
Point 1: It's better to discuss the role of trust in institutions, scientists, industry, environmental groups and people in different countries, e.g. , developing countries and developed countries.
Response: Thank you for the suggestion. In the revised version of the manuscript, we have better clarified the role of trust in different countries, e.g. , developing countries and developed countries (RL 150-165):
“In light of the preceding considerations and the pivotal significance of trust in the context of climate change adaptation, it is imperative to highlight the variegated psychological trust dynamics within communities, institutions, and scientific spheres, particularly as they differ between developing and developed nations. Within developed industrialized societies, individuals engage in a broader spectrum of societal affiliations, wherein dependence on community trust is notably diminished, supplanted instead by trust in institutional mechanisms overseeing communal governance on a larger scale. Conversely, in developing countries where localized communities wield substantial influence over individual livelihoods, the bedrock of social capital primarily rests upon community trust. Nevertheless, in the absence of complementary forms of trust, this reliance on community trust may foster insular, tightly cohesive communities, hindering broader involvement with developmental perspectives, efficiency, and innovation—including trust in scientific and technological domains—across more expansive, diverse groups. Within this framework, developing nations, characterized by less developed formal institutional infrastructures, are characterized by lower levels of institutional trust compared to their developed counterparts30.”
Point 2: Takeaway for Practice is also encouraged to be included in this paper. It is better to be clear enough to present your policy recommendations for local practice.
Response: Thank you for the suggestion. In the revised version of the manuscript, we have added practical implications emerged from the review, as follows (RL 710-735):
“With respect to this, this review yields several practical implications. The results emphasize the importance of improving risk communication between governmental bodies and the public. Specifically, local authorities and experts are required to facilitate the distribution of risk-related information and enhance the safety standards of a region through a participatory, bottom-up approach. However, effective communication concerning environmental hazards entails more than merely informing citizens; it requires an assessment of their understanding of the messages conveyed. Those engaged in communication must be aware of the community's needs, preferences, level of comprehension, and favored communication channels. Highlighting the capability, willingness, and entitlement of individuals to actively engage in CCA demonstrates how professionals can benefit from their involvement. Additionally, given current climate forecasts, community participation in practical training such as civil protection exercises became increasingly essential to improve prevention, preparedness, and response to natural disasters. Moreover, comprehending individuals' perceptions of natural hazard risks is critical for effectively disseminating information. The ultimate goal is to cultivate trust in authorities and enhance the ability to respond to significantly impactful events, thereby strengthening community resilience. Identifying the needs of citizens constitutes the initial and indispensable step toward promoting the exchange of intervention policies aimed at fortifying resilience and adaptation to CCA.”

Reviewer 3 Report
Comments and Suggestions for Authors
Dear authors, it is a pleasure to greet you, considering your article entitled: Psychological trust dynamics in climate change adaptation decision-making processes: A literature review. I have read and reviewed the article proposed for the Journal Sustainability.
I consider that the article is interesting and provides important contributions to improve the understanding of psychological aspects of human trust considering the actions that are being implemented in the world to achieve adaptation to climate change. Where the most relevant contribution is, without a doubt, the contribution to the understanding of the dynamics of psychological trust in the decision-making processes where different actors are part, related to the implementation of adaptation measures to climate change that are being carried out to stop the global climate crisis.
As a reviewer I can indicate that the article has a state of major revisions, and to be published it must substantially improve several aspects and respond to the following observations. Below, I communicate my impressions and recommendations to further increase the quality of the document submitted for peer review:
1.- Please allow me to mention that there is a confusion of basic elementary concepts in the abstract and in several chapters of the manuscript that need to be analyzed, revised, and corrected by the authors. These concepts are about the definition of adaptation to climate change and mitigation to climate change. Adaptation to climate change is defined as all those actions or measures that allow human beings to increase their resilience capacity, to face the impacts of climate change, manifested by extreme climate events, and thus finally, avoid damage (for example, works engineering to prevent disasters, territorial planning plans based on risk reduction, and nature-based solutions to avoid damage from extreme hydrometeorological events, among others). On the other hand, climate change mitigation is defined as all those actions or measures that allow humans to confront and appease the cause of climate change that corresponds to the generation of greenhouse gases, and thus finally, reduce the global temperature of the planet (for example, closure of electric power plants from fossil sources, insertion of electric power plants from renewable sources, and promotion of energy efficiency policies, among others).
2.- The abstract of the manuscript does not adequately describe the idea to be developed in the literature review on the topic stated in the title. The writing needs to be improved, since the context of the topic, the methodology to be implemented and the contribution of the review are not entirely clear. In particular, the methodology used should be better defined in the abstract, indicating details of the validated method considered. In this sense, a maximum length of 250 words is suggested. Please consider the following structure: (i) introduction and context, (ii) brief description of the validated methodology implemented and (iii) brief description of the main findings of the research. Please consider that the current version of the abstract where the results are mentioned states that: “The results reveal that trust dynamics influence different communities adaptive actions linked to renewable energy technologies adoption, low- and high-impact mitigation behaviors, and paying taxes for actions that mitigate climate change.” In this text, there is a conceptual error and confusion between adaptation to climate change and mitigation of climate change, please correct it. Finally, if “Psychological trust dynamics” is mentioned in the title, I think it is pertinent that those three words be mentioned in the abstract.
3.- In the keywords presented, there are repeated words. It is suggested to indicate the following keywords: Psychological trust dynamics; climate change adaptation; communities; government; Institutions; decision-making processes;
4.- The format of the citations presented in the manuscript is not correct according to the MDPI guidelines in its Sustainability Journal (Numbers should not be configured as superscripts), please review other articles already published and correct.
5.- In the introduction of the article specifically in: “1.1. Trust, climate change adaptation, and climate change decision-making”, after the presentation of the global problem of climate change and the vulnerability that exists in many regions susceptible to its impacts, it is necessary to write about the context of practical cases in the world on the implementation of adaptation measures to climate change, pointing out how different experiences in the world are being successful through resilience. Please do not only consider cases from Europe, but, for example, consider cases from Asia and South America. In that sense, consider the following articles as examples:
• Zhou, K.; Wang, S.; Feng, Y. How Is Spatial Planning Adapting to Climate Change? A Textual Analysis Based on the Territorial and Spatial Plans of 368 Chinese Cities. Land 2023, 12, 1993.
• Cacciuttolo, C.; Garrido, F.; Painenao, D.; Sotil, A. Evaluation of the Use of Permeable Interlocking Concrete Pavement in Chile: Urban Infrastructure Solution for Adaptation and Mitigation against Climate Change. Water 2023, 15, 4219.
6.- It is suggested to modify the following title: “1.1. Trust, climate change adaptation, and climate change decision-making”, by: 1.1 Psychological trust dynamics, climate change adaptation, and climate change decision-making Process.
7.- What is the main research question addressed by the article? Please provide this information in the final part of the introduction chapter.
8.- The chapter called: “1.3. Research Objectives” objectives should be reviewed and reformulated: Please remember that the different objectives should not repeat their verbs in the infinitive, have only one verb in the sentence and describe in a clear, concise and precise way the goal to be achieved in the research ( Consider the following as examples of verbs that can apply to this review: identify, characterize, etc.). In this sense, specific objective 1 must be reformulated and it is suggested to modify the wording of specific objective 2 as follows: “Understand how psychological trust dynamics are linked to climate change decision-making processes among communities, governments and institutions.”
9.- It is necessary to make a substantial improvement to the methodology chapter, unfortunately the methodology chapter presented is very weak. The methodology of a research is a central and important part, which is why in this chapter the authors must clearly define the following in subchapters: (i) research materials and resources considered, (ii) methodological procedure used, and (iii) assumptions and criteria considered. In that sense, it is suggested to consider the use of some validated methodology for literature review research. It is recommended to consider the method called PRISMA, which stands for: Preferred Reporting Items for Systematic Reviews and Meta-Analyses (PRISMA) guidelines. It is suggested to review the details of this methodology according to the following publication:
· Oláh, J.; Krisán, E.; Kiss, A.; Lakner, Z.; Popp, J. PRISMA Statement for Reporting Literature Searches in Systematic Reviews of the Bioethanol Sector. Energies 2020, 13, 2323
10.- In the reformulated Methodology chapter please provide information about the advantages and disadvantages regarding the presented Methodology considering this Research topic.
11.- It is suggested in the results chapter to show not only texts or tables, but it would be very interesting considering the literature review carried out on different cases in different countries of the world, to show: (i) diagrams or schemes that show the relationship of the research variables considered, and (ii) graphs with the distribution of the publications found on the research topic considering each country. To do this, it is recommended to use the free access tool called Datawrapper.
12.- In the chapter called discussion of results, a validation of the results obtained must be carried out considering a comparison with results obtained by other researchers in similar research. This means citing and referencing literature related to “Psychological trust dynamics in climate change adaptation decision.” “making processes: A literature review”. Furthermore, in the current version of the manuscript the following chapter “6. Limitations and future directions” should be part of the discussion of results as two subchapters, one called limitations and another subchapter called future directions, please correct.
13.- Please answer the following question with appropriate information in the discusión chapter: What does it add to the subject area compared with other published material?
14.- Please describe how the conclusions are or are not consistent with the evidence and arguments presented. Please also indicate if all main questions posed were addressed.
15.- The article uses many abbreviations. Therefore, it is necessary to generate a subchapter called a list of abbreviations with a table, a subchapter that must be located after the subchapter called conflicts of interest and before the references chapter. Consider the following example:
Abbreviations
|
ICTs |
Information and Communication Technologies |
|
CAPEX |
Capital Costs |
|
OPEX |
Operational Costs |
|
masl |
Meters above sea level |
16.- The article presented has a high similarity index equivalent to approximately 45%. Please review the writing of your manuscript and apply the technique of paraphrasing the texts in order to reduce the similarity index as much as possible.
17.- A review of the English in the article by a native english person is suggested.
Kind regards,
Comments on the Quality of English LanguageModerate editing of English language required
Author Response
Response to Reviewer 3 Comments
Dear authors, it is a pleasure to greet you, considering your article entitled: Psychological trust dynamics in climate change adaptation decision-making processes: A literature review. I have read and reviewed the article proposed for the Journal Sustainability.
I consider that the article is interesting and provides important contributions to improve the understanding of psychological aspects of human trust considering the actions that are being implemented in the world to achieve adaptation to climate change. Where the most relevant contribution is, without a doubt, the contribution to the understanding of the dynamics of psychological trust in the decision-making processes where different actors are part, related to the implementation of adaptation measures to climate change that are being carried out to stop the global climate crisis.
As a reviewer I can indicate that the article has a state of major revisions, and to be published it must substantially improve several aspects and respond to the following observations. Below, I communicate my impressions and recommendations to further increase the quality of the document submitted for peer review:
Point 1: Please allow me to mention that there is a confusion of basic elementary concepts in the abstract and in several chapters of the manuscript that need to be analyzed, revised, and corrected by the authors. These concepts are about the definition of adaptation to climate change and mitigation to climate change. Adaptation to climate change is defined as all those actions or measures that allow human beings to increase their resilience capacity, to face the impacts of climate change, manifested by extreme climate events, and thus finally, avoid damage (for example, works engineering to prevent disasters, territorial planning plans based on risk reduction, and nature-based solutions to avoid damage from extreme hydrometeorological events, among others). On the other hand, climate change mitigation is defined as all those actions or measures that allow humans to confront and appease the cause of climate change that corresponds to the generation of greenhouse gases, and thus finally, reduce the global temperature of the planet (for example, closure of electric power plants from fossil sources, insertion of electric power plants from renewable sources, and promotion of energy efficiency policies, among others).
Response: Thank you for the suggestion. We have revised the whole manuscript based on these the suggestions.
Point 2: The abstract of the manuscript does not adequately describe the idea to be developed in the literature review on the topic stated in the title. The writing needs to be improved, since the context of the topic, the methodology to be implemented and the contribution of the review are not entirely clear. In particular, the methodology used should be better defined in the abstract, indicating details of the validated method considered. In this sense, a maximum length of 250 words is suggested. Please consider the following structure: (i) introduction and context, (ii) brief description of the validated methodology implemented and (iii) brief description of the main findings of the research. Please consider that the current version of the abstract where the results are mentioned states that: “The results reveal that trust dynamics influence different communities adaptive actions linked to renewable energy technologies adoption, low- and high-impact mitigation behaviors, and paying taxes for actions that mitigate climate change.” In this text, there is a conceptual error and confusion between adaptation to climate change and mitigation of climate change, please correct it. Finally, if “Psychological trust dynamics” is mentioned in the title, I think it is pertinent that those three words be mentioned in the abstract.
Response: Thank you for the suggestion. We have revised the whole abstract based on these the suggestions, as follows (RL 11-26):
“There has been a growth in interest among academics and professionals in psychological trust dynamics during climate change adaptation. This literature review aimed to examine the research related the implications of the psychological trust dynamics in climate change adaptation from different levels of analysis, looking at the different phases of adaptation and considering its importance in climate change decision-making. The method consists of systematically reviewing psychological trust dynamics researches through scientific journal articles on the topic using appropriate and relevant search keywords (e.g., trust, community, natural hazard, climate change adaptation, decision-making). A total of 25 studies met inclusion criteria, and all selected papers focused only on the last two phases of the whole climate change adaptation cycle. The results reveal that psychological trust dynamics influence different communities' adaptive actions linked to renewable energy technologies adoption, and low- and high-impact mitigation behaviors, such as paying taxes for actions that mitigate climate change. This literature review further demonstrates that psychological trust dynamics holds significant way over individuals' decision-making regarding climate change adaptation and mitigation, thereby resulting in the most immediate or direct consequences for climate systems. Therefore, it is essential to actively promote the culture of trust within the context of adapting to climate change”
Point 3: In the keywords presented, there are repeated words. It is suggested to indicate the following keywords: Psychological trust dynamics; climate change adaptation; communities; government; Institutions; decision-making processes;
Thank you for the suggestion. We have modified keywords using only terms that best matches the main theme of the research (“trust; climate change adaptation; climate-related hazards; climate change decision-making”).
Point 4: The format of the citations presented in the manuscript is not correct according to the MDPI guidelines in its Sustainability Journal (Numbers should not be configured as superscripts), please review other articles already published and correct.
Response: Thank you for the suggestion. We have corrected the format.
Point 5: In the introduction of the article specifically in: “1.1. Trust, climate change adaptation, and climate change decision-making”, after the presentation of the global problem of climate change and the vulnerability that exists in many regions susceptible to its impacts, it is necessary to write about the context of practical cases in the world on the implementation of adaptation measures to climate change, pointing out how different experiences in the world are being successful through resilience. Please do not only consider cases from Europe, but, for example, consider cases from Asia and South America. In that sense, consider the following articles as examples:
- Zhou, K.; Wang, S.; Feng, Y. How Is Spatial Planning Adapting to Climate Change? A Textual Analysis Based on the Territorial and Spatial Plans of 368 Chinese Cities. Land 2023, 12, 1993.
- Cacciuttolo, C.; Garrido, F.; Painenao, D.; Sotil, A. Evaluation of the Use of Permeable Interlocking Concrete Pavement in Chile: Urban Infrastructure Solution for Adaptation and Mitigation against Climate Change. Water 2023, 15, 4219.
Response: Thank you for the suggestion. We've included information about real-world case studies demonstrating the implementation of measures to adapt to climate change, as follows (RL 230-249):
“Recent studies in global literature underscore the pivotal role that cities play in tackling climate change by both mitigating its effects and adapting to them. Take, for instance, spatial planning, which is increasingly acknowledged as a fundamental tool. It facilitates the integration of urban design, optimization of spatial structure, and efficient management of land use and infrastructure. Moreover, it acts as a governance mechanism at the local level, shaping policy measures for spatial development. In recent years, western nations have incorporated climate adaptation objectives into their spatial planning policies, while eastern countries have been involved in climate programs since the late 1980s. Recent revisions in territorial and spatial planning aim to holistically coordinate various spatial aspects. Nevertheless, there's a pressing need to strengthen the role of territorial planning in addressing climate change at the urban level, especially due to inadequate awareness, limited analytical capability, and insufficient action regarding climate change issues. Generally, larger cities, with their robust economies and dense populations, tend to excel in climate change initiatives compared to smaller ones40,41.”
Point 6: It is suggested to modify the following title: “1.1. Trust, climate change adaptation, and climate change decision-making”, by: 1.1 Psychological trust dynamics, climate change adaptation, and climate change decision-making Process.
Response: Thank you for the suggestion. We have changed the title (RL 164-165).
Point 7: What is the main research question addressed by the article? Please provide this information in the final part of the introduction chapter.
Response: Thank you for the suggestion. We have added the main research question addressed by the article in the final part of the introduction chapter, as follows (RL 316-321):
“For these reasons, we conducted a literature review examining the implications of trust dynamics in climate change decision-making. The aim of this literature review is to analyze the impact of trust on decision-making processes in the context of adaptation to climate change. On this basis, the section of results will endeavor to provide further insights into the role of trust in CCA processes and climate change decision-making.”
Point 8: The chapter called: “1.3. Research Objectives” objectives should be reviewed and reformulated: Please remember that the different objectives should not repeat their verbs in the infinitive, have only one verb in the sentence and describe in a clear, concise and precise way the goal to be achieved in the research (Consider the following as examples of verbs that can apply to this review: identify, characterize, etc.). In this sense, specific objective 1 must be reformulated and it is suggested to modify the wording of specific objective 2 as follows: “Understand how psychological trust dynamics are linked to climate change decision-making processes among communities, governments and institutions.”
Response: Thank you for the suggestion. In the revised version of the manuscript, we have better clarified the research objectives behind the review (RL 323-334):
“We conducted a literature review examining the implications of trust dynamics in climate change decision-making. The purpose was to analyze the outcomes of pertinent studies, systematize the deriving knowledge, and identify potential gaps in the existing literature, so to identify future directions for improving research and practice on decision making and development of trust in in the context of adaptation to climate change.
Specifically, the present literature review was conducted to:
1) Analyze how psychological trust dynamics affects the different phases of CCA: preparing the ground for adaptation, assessing climate change risks and vulnerabilities, identifying adaptation options, assessing and selecting adaptation options, implementing adaptation, and monitoring and evaluating adaptation.
2) Analyze how psychological trust dynamics are linked to climate change decision-making processes among communities, governments and institutions.”
Point 9: It is necessary to make a substantial improvement to the methodology chapter, unfortunately the methodology chapter presented is very weak. The methodology of a research is a central and important part, which is why in this chapter the authors must clearly define the following in subchapters: (i) research materials and resources considered, (ii) methodological procedure used, and (iii) assumptions and criteria considered. In that sense, it is suggested to consider the use of some validated methodology for literature review research. It is recommended to consider the method called PRISMA, which stands for: Preferred Reporting Items for Systematic Reviews and Meta-Analyses (PRISMA) guidelines. It is suggested to review the details of this methodology according to the following publication:
- Oláh, J.; Krisán, E.; Kiss, A.; Lakner, Z.; Popp, J. PRISMA Statement for Reporting Literature Searches in Systematic Reviews of the Bioethanol Sector. Energies 2020, 13, 2323
Response: Thank you for the suggestion. In the revised version of the manuscript, we have better clarified the methods section behind the review (RL 337-357):
“This literature review has been carried out in order to provide a comprehensive understanding of trust dynamics in the realms of CCA. In the preliminary phase, we systematically examined contemporary literature pertaining to the subject matter with the aim of identifying investigations employing rigorous methodologies. Then, we selected the appropriate and relevant search keywords (e.g., trust, community, natural hazard, climate change adaptation, decision-making). This methodological approach sought to enrich and contemporize the knowledge repository within this discipline, thereby securing its pertinence for both present and future cohorts. The selection of keywords was undertaken to encompass diverse dimensions of CCA and trust, spanning across community and institutional spheres. Subsequently, we performed searches in academic databases (PubMed/Medline, ISI Web of Science, and SCOPUS). Empirical studies that investigated the role of trust dynamics in the realms of CCA were deemed eligible based on the following inclusion criteria: (1) they employed cohort, case-control, cross-sectional study, and/or experimental design. Publications were excluded if (1) they were not original articles (e.g., proceeding, review, opinion paper, or dissertation), and (2) they did not specifically focus on natural disaster. After conducting a thorough scrutiny of the retrieved literature, we examined the studies pertaining to the topic. The strength of utilizing this methodology resides in its thorough integration of scientific articles that pertain to the central topic under examination in this literature review. Conversely, its drawback is the omission of grey literature, wherein significant studies might exist that could facilitate a more profound understanding of the phenomenon.”
Point 10: In the reformulated Methodology chapter please provide information about the advantages and disadvantages regarding the presented Methodology considering this Research topic.
Response: Thank you for the suggestion. In the revised version of the manuscript, we have better clarified also the advantages and disadvantages regarding the presented methodology, as follow (RL 353-357):
“The strength of utilizing this methodology resides in its thorough integration of scientific articles that pertain to the central topic under examination in this literature review. Conversely, its drawback is the omission of grey literature, wherein significant studies might exist that could facilitate a more profound understanding of the phenomenon”.
Point 11: It is suggested in the results chapter to show not only texts or tables, but it would be very interesting considering the literature review carried out on different cases in different countries of the world, to show: (i) diagrams or schemes that show the relationship of the research variables considered, and (ii) graphs with the distribution of the publications found on the research topic considering each country. To do this, it is recommended to use the free access tool called Datawrapper.
Response: Thank you for the suggestion. We have added a graph with the distribution of the publications found on the research topic considering each country.
Point 12: In the chapter called discussion of results, a validation of the results obtained must be carried out considering a comparison with results obtained by other researchers in similar research. This means citing and referencing literature related to “Psychological trust dynamics in climate change adaptation decision.” “making processes: A literature review”. Furthermore, in the current version of the manuscript the following chapter “6. Limitations and future directions” should be part of the discussion of results as two subchapters, one called limitations and another subchapter called future directions, please correct.
Response: Thank you for the suggestion. We have modified the final part of the manuscript as you suggested.
Point 13: Please answer the following question with appropriate information in the discusión chapter: What does it add to the subject area compared with other published material?
Response: Thank you for the suggestion. In the revised version of the manuscript, we have better clarified also what this review adds to previous results, as follow (RL 729-736):
“The results of this review furnish a comprehensive and methodically organized overview of the importance attributed to psychological trust dynamics in the domain of climate change. Preceding this review, the implications stemming from psychological trust dynamics had not undergone thorough scrutiny throughout the entirety of the CCA cycle, leading to a deficit in the integration of studies within this conceptual framework. Consequently, this review stands as a pivotal point of reference for scholars, prompting the initiating new investigations into all facets of the complete cycle where the dynamics of psychological trust have yet to be thoroughly investigated.”
Point 14: Please describe how the conclusions are or are not consistent with the evidence and arguments presented. Please also indicate if all main questions posed were addressed.
Response: Thank you for the suggestion. We have clarified how the conclusions are consistent with the evidence and arguments presented, as follow (RL 748-754):
“The present review facilitated the addressing of research objectives and delineated affirmative associations between psychological trust dynamics and fundamental constituents of decision-making pertaining to climate change. Results of the review provides evidence concerning the effects of psychological trust dynamics in CCA and in individuals’ climate change decision-making: both institutional, community, and scientifical-technological trust are associated to climate change decision-making in the CCA cycle.”
Point 15: The article uses many abbreviations. Therefore, it is necessary to generate a subchapter called a list of abbreviations with a table, a subchapter that must be located after the subchapter called conflicts of interest and before the references chapter. Consider the following example:
Abbreviations
- ICTs: Information and Communication Technologies
- CAPEX: Capital Costs
- OPEX: Operational Costs
- Masl: Meters above sea level
Response: Thank you for the suggestion. We have added a list of abbreviations.
Point 16: The article presented has a high similarity index equivalent to approximately 45%. Please review the writing of your manuscript and apply the technique of paraphrasing the texts in order to reduce the similarity index as much as possible.
Response: Thank you for the suggestion. We have reviewed the manuscript and paraphrased the texts in order to reduce the similarity index as much as possible.
Point 17: A review of the English in the article by a native english person is suggested.
Response: Thank you for the suggestion. We have reviewed English language.

Reviewer 4 Report
Comments and Suggestions for Authors
Please incorporate additional papers from 2023. The author noted, "A total of 25 studies met inclusion criteria, and all 16 selected papers focused solely on the final two phases of the entire climate change adaptation cycle." What are the reasons for this exclusive focus on the last two phases? Please provide a detailed discussion of the strengths and weaknesses of quantitative research design and mixed methods research design. Please provide detailed information on the distribution of papers by gender, age, and race. Please provide further explanation on the statement, "In summary, we found that trust is also related to proactive behaviors within the field of climate change adaptation, particularly during the monitoring and evaluation phase."
Author Response
Response to Reviewer 4 Comments
Please incorporate additional papers from 2023. The author noted, "A total of 25 studies met inclusion criteria, and all 16 selected papers focused solely on the final two phases of the entire climate change adaptation cycle." What are the reasons for this exclusive focus on the last two phases? Please provide a detailed discussion of the strengths and weaknesses of quantitative research design and mixed methods research design. Please provide detailed information on the distribution of papers by gender, age, and race. Please provide further explanation on the statement, "In summary, we found that trust is also related to proactive behaviors within the field of climate change adaptation, particularly during the monitoring and evaluation phase."
Response: Thank you for the suggestion. We have thoroughly examined the entire manuscript and implemented revisions in accordance with all suggestions provided by the reviewer, with the exception of incorporating additional literature specific to the year 2023. Despite conducting a new search, no studies deemed pertinent to the objectives of our review were identified. Consequently, we are confident that the manuscript has undergone substantial enhancement as a result of these revisions. All corrections have been clearly delineated to facilitate the reviewer's assessment (RL 367-381; 389-405).

Round 2
Reviewer 3 Report
Comments and Suggestions for Authors
The authors have made positive changes to the manuscript improving the quality of it. Finally, I recommend acceptance of the article.
Regards,
Comments on the Quality of English LanguageModerate english review is required.